# Synthesis of MnO/C/NiO-Doped Porous Multiphasic Composites for Lithium-Ion Batteries by Biomineralized Mn Oxides from Engineered *Pseudomonas putida* Cells

**DOI:** 10.3390/nano11020361

**Published:** 2021-02-01

**Authors:** Jin Liu, Tong Gu, Li Li, Lin Li

**Affiliations:** State Key Laboratory of Agricultural Microbiology, Huazhong Agricultural University, Wuhan 430070, China; liujinlj1987@163.com (J.L.); gutong1117@webmail.hzau.edu.cn (T.G.); lily999wy@126.com (L.L.)

**Keywords:** biogenic manganese oxides, lithium-ion battery, biotemplate, bacterial cell surface display, hollow porous biocomposite

## Abstract

A biotemplated cation-incoporating method based on bacterial cell-surface display technology and biogenic Mn oxide mineralization process was developed to fabricate Mn-based multiphasic composites as anodes for Li-ion batteries. The engineered *Pseudomonas putida* MB285 cells with surface-immobilized multicopper oxidase serve as nucleation centers in the Mn oxide biomineralization process, and the Mn oxides act as a settler for incorporating Ni ions to form aggregates in this process. The assays using X-ray photoelectron spectroscopy, phase compositions, and fine structures verified that the resulting material MnO/C/NiO (CMB-Ni) was porous multiphasic composites with spherical and porous nanostructures. The electrochemical properties of materials were improved in the presence of NiO. The reversible discharge capacity of CMB-Ni remained at 352.92 mAh g^−1^ after 200 cycles at 0.1 A g^−1^ current density. In particular, the coulombic efficiency was approximately 100% after the second cycle for CMB-Ni.

## 1. Introduction

The electrochemical performance of lithium-ion batteries depends largely on the properties of the electrode materials. There have been many reports on materials with high lithium storage capacity, including transition metal oxides and metalloids, such as NiO, Co_3_O_4_, and ZnO [1,2,3]. Among the various available transition metal oxides, MnO has attracted significant attention due to its low cost, high theoretical capacity (756 mAh g^−1^), and relatively low running potential (1.032 V vs. Li/Li^+^) [1,4,5]. However, despite the promising electrochemical properties of MnO, its application as an electrode material is generally limited by poor rate discharge performance and short cycle life, which is caused by severe particle aggregation and a huge volume change in the charging-discharging process [6].

Two technical routes are presumable to solve this problem. The first route is to incorporate with other elements, particularly metal ions [7,8], nitrogen and oxides [9], to improve the stability of the crystal structure and to enhance the electrical contact between oxide particles, as well as between oxide particles and the current collector, as demonstrated by Li et al. who prepared Fe-Mn-O composites with different Fe/Mn ratios by a simple coprecipitation process. The resultant composites showed better electrochemical properties than those of single Fe_3_O_4_ or MnO electrode materials [10]. The second route for improving the electrode performance of MnO is to fabricate biocomposites with distinctive morphologies, such as microspheric, hollow, porous, and nuclear shell structures, which can enhance the contact between electrolyte and internal active substances to accelerate the transfer of Li^+^. Among the various available approaches for this purpose, the biotemplate and surface display technologies have increasing appeal due to the effectiveness, regeneration, and good performance of the methods available that allow the fabricated materials to be endowed with the complicated fine structures and functional characteristics of template organisms, as validated by several previously developed virus surface display systems, such as in phage M13 and other filamentous bacteriophages [11,12]. However, while the feasibility of combining the phage surface display technology and biotemplate method has been intensively investigated, there are still only a few studies on the application of other microbial surface display technologies that integrate a biotemplate process.

The key to biomineralization is to realize the ordered sedimentation, growth, and assembly of metal oxides under the regulation and control of organic matter. With the revelation of the mineralization mechanism, the mineralization reconstruction of specific functional organisms via generation of functional inorganic shells or crystals by biomineralization has shown potential for the fabrication of metal oxide materials. Previous research demonstrated that the surface display of Mn oxidases on bacterial cells can retain their biological activity by fusion of an activity-independent amino acid to a cell wall-binding anchor protein, which further enables fast and goal-oriented Mn oxidation reaction and formation of Mn oxide deposit layers on cell surface [13]. Due to the non-uniformity of Mn valence states and the presence of lattice vacancies in biogenic Mn oxides, additional cations were required to compensate the negative charge. Moreover, biogenic Mn oxides have the characteristics of settlers for metal ions, the additional metal ions, such as Co^2+^, Ni^2+^, Zn^2+^, and Pb^2+^, could be utilized as mineral components through a redox reaction in the crystal growth process of Mn oxides [14,15,16]. This characteristic suggested a kind of technical route to realize facile fabrication of multi-metal mineral materials under mild conditions. The minerals were released in a way that the Mn oxidases do not become encumbered by Mn oxides, which are less likely to associate with the enzyme [17]. The dense aggregates, consisting of Mn oxides and bacteria, could be formed by gradual sedimentation [13]. The carbonaceous organic compounds arising from bacteria could be used as biological templates to provide a fundamental structure for subsequent materials. The organic carbon also could be decomposed in high-temperature annealing to form carbon, which form a reducing atmosphere to transform MnO_2_ to MnO as well as served as a carbon matrix for Mn oxides embedding after calcination. The space formed in the decomposition process led to a special morphology with multilevel micro-nano structures of porous irregular microspheres. The Mn oxides produced in the mineralization were used as the source for MnO_x_ after calcination.

In this study, a porous MnO/C/NiO composite (CMB-Ni) was prepared using bacterial surface display technology and a biotemplate method (Scheme 1). *Pseudomonas putida* MB285 is an engineered strain expressing surface-immobilized multicopper oxidase (MCO) which can catalyze the oxidization of Mn^2+^ [13,18,19] and gradually forming the microspherical porous aggregates with biofilm coated strains and Mn oxides. A unique incorporating method by biomineralization was designed, in which Ni^2+^ ions was incorporated into the crystal structure of biogenic Mn oxides under mild conditions. The aggregate precursor was subsequently carbonized at different high temperatures and carbonaceous MnO-based multiphasic nanocomposites were fabricated. The crystallization state evolution of the resultant materials was further determined. The electrochemical properties of nanocomposites as the anode materials for lithium-ion batteries were measured. The possible reasons for performance differences of the materials are discussed.

## 2. Materials and Methods

### 2.1. Chemicals, Bacterial Strains, Culture Conditions, and Mn^2+^-Oxidizing Activity Assay

All chemicals were of analytical grade and were used as received without further purification. NiCl_2_.6H_2_O (≥99%) was purchased from Sigma-Aldrich Co. (LLC., Shanghai, China) and was used as the Ni ion source in the Leptothrix medium [20] for the cation-incorporated composite.

A previously constructed engineered bacterial strain *P. putida* MB285, which expresses the recombinant “(InaQ-N)_3_-WlacD”, a fusion protein with three tandem repeated InaQ-N anchors and a mutated bacterial laccase WlacD, was used as the host strain for the production of Mn oxide aggregates under laboratory shaker-flask trials [18]. The strain was routinely grown in liquid Leptothrix medium supplemented by 1 mM MnCl_2_ at 28 °C at a shaking of 150 rpm. The Mn^2+^-oxidizing activity was measured over a time course of 15 d. A standard leucoberbelin blue (LBB) spectrophotometry assay was performed for Mn oxide quantification, as previously described [21].

### 2.2. Preparation of MnO/C/NiOComposite

The preparation and use flow of the MnO/C/NiO porous composite is illustrated in Scheme 1. Firstly, for the preparation of Mn oxide aggregates, the suspension cultures were harvested after cultivating *P. putida* MB285 cells for 12 d. The yielded biogenic MnO_2_/bacteria composites (BMB) were rinsed three times with double distilled water and then dried at 80 °C for 10 h. The biogenic MnO_2_/bacteria Ni-incorporated aggregates (BMB-Ni) were prepared in parallel, except that NiCl_2_ was added to the culture at a final concentration of 1 mM 24 h after the strain was inoculated. Secondly, for the thermochemical process, the MnO/C composite was obtained by calcination of the BMB in a tube furnace at 800 °C (temperatures in the range of 400−800 °C were investigated, the corresponding materials were defined as B400, B600 and B800, respectively) for 4 h with a ramp of 5 °C min^−1^ under a Ar atmosphere. The MnO/C/NiO composite (CMB-Ni) was prepared by calcination of BMB-Ni in a tube furnace at 800 °C.

### 2.3. Material Characterization

The phase purity and properties of the samples were measured using X-ray powder diffraction (XRD, Bruker D8 Advance diffractometer with Cu Kα radiation, Bruker, Karlsruhe, Germany) and X-ray photoelectron spectroscopy (XPS, Multilab 2000, Thermo Electron Corporation, London, UK) with monochromatic Mg Kα radiation (1253.6 eV). XPS analysis of the obtained materials was conducted on a VG Multilab2000 X-ray photoelectron spectrometer with an Al Kα X-ray source (1486 eV) and a base pressure of 3 × 10^−9^ Torr in the analytical chamber [22]. The charge effect was corrected by adjusting the binding energy of adventitious C (1 s) to 284.8 eV. The Shirley-type background was subtracted before deconvolution and data fitting using the parameters used by Nesbitt et al. [23]. For the multiplet peaks of Mn (2p_3/2_) for spectral fitting, a 20:80 ratio of the Lorentzian–Gaussian mix-sum function was used for all data fittings [24]. Scanning electron microscopy (SEM, Hitachi S-4700) and high-resolution transmission electron microscopy (HRTEM, JEM-2100F) equipped with an energy-dispersive spectroscopy (EDS) detector were used to investigate the microstructure of the sample and analyze its elements, the HRTEM was operated at an acceleration voltage of 200 kV. The surface functional groups were analyzed by a Fourier transform infrared spectrometer (FT-IR, VERTEX 70, Bruker, Karlsruhe, Germany). Thermogravimetric analysis (TGA, thermal analysis system, TG 209, Netzsch, Bavaria, Germany) was carried out in a N_2_ atmosphere from room temperature (RT) to 800 °C at a heating temperature speed of approximately 10 °C min^−1^. Raman spectra were obtained on a Renishaw InVia Raman spectrometer under a backscattering geometry (λ = 532 nm). Nitrogen adsorption-desorption was determined by Brunauer−Emmett−Teller (BET) tests using a Quantachrome Autosorb-1, JEDL-6390/LV, (JEOL, Tokyo, Japan).

### 2.4. Assays of Electrochemical Properties

The electrochemical performance of materials was assayed using 2016 coin-type cells with lithium as the counter electrode at RT. The working electrode was fabricated by mixing the active material, acetylene black, and a polyvinylidene fluoride binder in a weight ratio of 8:1:1. Each 1 M LiPF_6_ in the ethylene carbonate-dimethyl carbonate (1:1 by volume) was used as the electrolyte of the coin cells, which were assembled in an argon-filled glove box with a microporous membrane (Celgard 2300) as a separator. The cells were galvanostatically cycled at various current densities and the charge/discharge capacity was performed in a fixed voltage window of 0.01 to 3.0 V versus Li/Li^+^ using a battery tester (CT2001A Land Battery Testing System, Wuhan, China). Cyclic voltammogram (CV) tests of the electrode were carried out by a CHI 660c electrochemical workstation (Chenhua, Shanghai, China) with a scan rate of 0.1 mV s^−1^ between 0.01–3 V.

## 3. Results and Discussion

### 3.1. Formation and Characterization of BMB-Ni Aggregates

Previous investigations have revealed that the MCOs of a variety of Mn^2+^-oxidizing bacteria were directly involved in Mn^2+^ oxidation of host cells [25,26]. These MCO-mediated oxidation reactions all appear to be related to the cell surface [13,27]. To investigate the Mn^2+^-oxidizing activity of the engineered *P. putida* MB285 cells expressing surface-immobilized MCOs, a continuous Mn^2+^-rich culture under laboratory shake-flask conditions was conducted. The Mn^2+^-oxidizing activity was observed and gradually increased (Figure 1a) after cultivating *P. putida* MB285 cells for 7 d. The aggregation of cells started after 12 d and microspherical aggregates were gradually formed (Figure 1b,c). The aggregates were irregular spherical secondary particles composed of flaky Mn oxides adhered on bacteria surfaces and dispersed spherical Mn oxides. It is worth mentioning that the aggregate structure was dense and could still maintained its morphology intact at a SEM operated voltage of 200 kv, which was consistent with that finding in other studies [13]. The aggregate had a relatively uniform size with a diameter of 20–30 µm (Figure 1b). Interestingly, the Mn^2+^-oxidizing activity of mixed bacterial suspension increased significantly after adding Ni^2+^ (BMB-Ni), compared to that of net *P. putida* cells (BMB). Previous reports had found that LBB could also react with Co (III) oxides, so similar reactions may occur between oxidized Ni and LBB, which then increased the value of the determined oxides [15].

There were porous structures on the surface of the microspherical aggregates. The analysis of the BET specific area and porosity (Figure 1d) indicated that the specific surface area of BMB was 20.935 m^2^ g^−1^. The–Barrett–Joyner–Halenda (BJH) pore size was mainly distributed at 20 and 33.97 nm, which is classed as a mesoporous material (0–50 nm). The adsorption-desorption isotherm loop was H3-type (inset in Figure 1d). Usually, the H3-type material contained flaky particulate filled polymers to form a slit structure [28], consistent with the SEM observation results. According to the composition of the surface element in aggregates analyzed by line-scan EDS (Figure 1e), the content of Mn was 7.78%, which was uniformly distributed on the microsphere surface. Meanwhile, the contents of N and O were 19.28% and 63.99% (Appendix A), respectively. This indicated that the main components in aggregates were organic biomass, consistent with the results of TGA (Appendix A). After heating under an Ar atmosphere, the mass loss of 8.9% below 100 °C was due to the volatilization of the attached water on the surface. The organic carbon could be decomposed in high-temperature annealing to form carbon, which forms a reducing atmosphere, so the mass loss of aggregates reached 59.9% when the temperature was varied from 100 to 800 °C, which was caused by the decomposition of organic matter and the reduction of MnO_2_ to MnO by a reducing atmosphere. Therefore, the content of organic matter from bacteria in BMB was 59.9%.

HRTEM was performed to further investigate the microstructure of BMB. As shown in Figure 2a,b, BMB had an irregular spherical structure. Many groups of Mn oxide nanocrystal particles were independently embedded in the organic matter produced by bacteria. The interplanar spacings of 0.279 and 0.362 nm corresponded to the *d* value of the (004) face of Na_0.7_MnO_2_ and the (012) face of MnCO_3_, respectively. The XRD analysis indicated that BMB was composed of four phases, i.e., Na_0.7_MnO_2_ (JCPDS card no. 27-0752), MnCO_3_ (JCPDS card no. 85-1109), CaCO_3_ (JCPDS card no. 17-0763) and Na_2_S_2_O_4_ (JCPDS card no. 72-1157) (Figure 2c). The diffraction peaks in the spectra were attributed to the (002), (004), (112), (023), (114) and (134) planes of Na_0.7_MnO_2_, the (012), (104), (110), (202), (024) and (116) planes of MnCO_3_, (220), (212) and (520) planes of CaCO_3_, and the (100), (−111), (020), (−323) and (−242) planes of Na_2_S_2_O_4_, respectively. The relatively weak strength of the diffraction peaks was caused by the high background of bioorganic matter in BMB. The presence of Na_2_S_2_O_4_ in four phases should be related to the S-containing components in the culture medium. Both Na_0.7_MnO_2_ and MnCO_3_ were from the Mn oxidation of BMB. The oxidation reaction from Mn^2+^ to Mn^4+^ actuated by MCO mainly produced Na_0.7_MnO_2_, while MnCO_3_ was the by-product under anaerobic conditions in continuous culture process due to the partial lack of oxygen [29,30].

The Mn^3+^ intermediate product existed in the electron transfer process of Mn oxidation. Since its oxidation was affected by the concentration of Mn, Mn oxides with different valence states were possibly generated [31,32]. Therefore, the proportion of Mn species in BMB was analyzed by XPS (Figure 2d). The spectral fitting results of Mn(2p_3/2_) spectra proved the presence of Mn^2+^, Mn^3+^, and Mn^4+^, with a proportion of 6.02, 24.62%, and 69.36%, respectively (Appendix A), indicating that the main oxidizing states of Mn ions were Mn^4+^, consistent with the XRD results. The FT-IR analysis showed that the Mn-O bonds were not detected for pure MB285 cells or the mixture of cells and MnCl_2_ without cultivation. However, after Mn oxidation of MB285 to produce BMB, a new vibration peak appeared at 612 cm^−1^, which was assigned to the Mn-O band (Appendix A) [33]. This further proved the formation of Mn oxide after the mineralization process.

### 3.2. Preparation and Characterization of MnO/C/NiO Hollow Porous Composite

Based on their mesoporous morphology and the feasibility of natural incorporate of metal ions in the Mn mineralization process, BMB aggregates were used as a biological template to design a facile modification method for electrode materials. Ni^2+^ was added into MB285 culture, the Ni^2+^-incorporated Mn oxide aggregates (BMB-Ni) were harvested and were used as the precursors for high-temperature annealing (800 °C). The organic carbons from organisms were decomposed in the annealing process and the reducing atmosphere was generated. Therefore, MnO_2_ was reduced to MnO. Meanwhile, the residual inorganic carbon was converted into a carbon coating and electroconductive carbon matrix. The space of organic matter formed cavities and macropores after decomposition. Thus, the porous MnO/C/NiO multiphasic composites were consequentially obtained after the annealing process.

In order to investigate the influence of the calcinations temperature on the surface morphology and internal structure of the materials, BMB was treated at 400 °C, 600 °C, and 800 °C. Calcinations treatments caused irregular porous particles for all materials, and the collapsed particles with a smallest volume were obviously observed for B800 (the material was treated at 800 °C) (Figure 3a and Appendix A). With the increase of calcination temperature, the surface roughness of the material decreased distinctly. A significant number of attachments disappeared and the pore size gradually increased with increasing calcination temperature (Figure 3b, Appendix A). The shrinkage of particle volume and the decrease of surface roughness could be caused by the decomposition of organic matter in the material. The EDS analysis of B800 indicated that the contents of Mn, C and O were 28.75%, 9.78% and 32.2%, respectively (Appendix A; Appendix A). The content of Mn was 3.69 times that in BMB (Figure 1e). The TGA assay of B800 and CMB-Ni (Appendix A) revealed that the carbon content was not more than 10%, which was consistent with the analysis by EDS, verifying the decomposition of organic matter at high temperatures. Figure 3c showed that many particle attachments existed on the surface of the CMB-Ni, but the modified material was still morphologically similar to B800, i.e., irregularly spherical with porous structures on the surface (Figure 3d). We speculated that these particle attachments were composed of Mn and incorporated elements after reaction at high temperatures. The particles were analyzed by EDS linear scan assays (Appendix A). The results indicated that the main elements in particles of CMB-Ni were Mn, O, C, Ni, P, and Ca.

The FT-IR analysis of B400, B600 and B800 showed that all three materials had absorption peaks at 561 and 1047 cm^−1^, which could be attributed to the vibration peaks of Mn–O and C–O, respectively (Figure 3e). This should be related to the presence of Mn oxides and carbonates in the materials. However, more vibration peaks were observed in B400 than B600 and B800, indicating the abundant chemical bonds. The peaks at 1434 and 1601 cm^−1^ could be attributed to the C=C and C=O bands, respectively. In addition, the vibration peak at 3390 cm^−1^, attributed to O–H, was also observed in B400 and B600. The C=C, C=O and O-H chemical bonds were a result of the biological functional groups. The presence of these bands proved that partial biological functional groups could be retained at a calcination temperature of 400 °C, while the biological functional groups were completely decomposed at 800 °C. The precursor BMB was composed of biogenic Mn oxides and organic carbon (Figure 2 and Appendix A). The excessive cation vacancies in biogenic Mn oxides and the presence of biological functional groups in organic carbon could lead to very low internal electronic conductivity. However, the metastable cation vacancies and biological functional groups could be removed by proper high temperature treatment. Thus, the organic carbon was converted to inorganic carbon with improved conductivity [34,35]. Raman spectra show two characteristic peaks in B800 at 1344.47 and 1580.75 cm^−1^, which correspond to the D- (disorder carbon) and G-bands (graphitic carbon), respectively. The peaks at 630.7 cm^−1^ are in accordance with the characteristic vibration mode of Mn_3_O_4_ rather than MnO_2_ or MnO, which mainly attributed to the local heating effect and photochemically induced transformations under beam irradiation (Appendix A) [1]. This result demonstrates the presence of Mn-O and consistent with the FT-IR data. According to the SEM, FT-IR and Raman spectra analysis results, the materials obtained at a carbonization temperature of 800 °C (B800 and CMB-Ni) exhibited better morphology and complete decomposition of biological functional groups, therefore, it was selected for subsequent studies.

The phase composition of the B800 and CMB-Ni, as well as the phase evolution after high-temperature annealing, was compared and analyzed by XRD (Figure 4a,b). The results showed that B800 contained four phases, i.e., MnO (JCPDS card no. 78-0424), MnCO_3_ (JCPDS card no. 85-1109), CaCO_3_ (JCPDS card no. 17-0763) and MnSO_4_ (JCPDS card no. 29-0898) (Figure 4a). The diffraction peaks in the spectra were attributed to the (111), (200), (220), (311) and (222) planes of MnO, the (012), (104) and (110) planes of MnCO_3_, the (110), (021), (111) and (042) planes of MnSO_4_, and the (220) and (221) planes of CaCO_3_. The diffraction peaks of some phases were overlapped, for example, (221) of CaCO_3_ and (104) of MnCO_3_, and (111) of MnSO_4_ and (012) of MnCO_3_. CMB-Ni (Figure 4b) also contained the four phases described above. However, in addition, new phases of NiO (JCPDS card no. 44-1159) and MnS (JCPDS card no. 06-0518) were observed in CMB-Ni. The new diffraction peaks corresponded to the (101), (110) and (202) planes of NiO, and the (200), (220) and (420) planes of MnS. The discrepancy of the phase compositions in B800 and CMB-Ni might be explained as follows. The reducing atmosphere was generated in the annealing process due to the decomposition of biological carbon. Then, Na_0.7_MnO_2_ in BMB was reduced to MnO. Na_2_S_2_O_4_ reacted with residual Mn^2+^ to form MnSO_4_. MnS was the common product in Mn oxidation when the oxygen supply was insufficient [29]. The components, including CaCO_3_ and MnCO_3_, remained. The XRD analysis of B400 and B600 also proved this conclusion (Appendix A). B400 was incompletely reduced to form Mn_2_O_3_ (JCPDS card no. 71-0635) under reducing atmosphere. Moreover, the background of the diffraction pattern was relatively high and the characteristic diffraction peaks were not obvious, indicating that the organic matter in the material was partially decomposed. The high background signal interference still existed. At higher temperatures, B600 and B800 were fully reduced to MnO. As a result, the characteristic diffraction peaks were clear and sharp. The good crystallization confirmation indicates the significant decrease of organic matter content, consistent with the FT-IR and TGA results for B800.

The microstructures of B800 and CMB-Ni were further analyzed by HRTEM. Under calcination conditions of 800 °C, the incorporating of metal ions did not change the overall structure of the material, which was still irregularly spherical (Figure 4c,e). In B800, the interplanar spacing of the main lattice image was 0.22 nm, which could be attributed to the (200) planes of MnO. However, a situation where three lattice fringes were stacked with each other was also observed. The lattice fringe with interplanar spacings of 0.254 and 0.22 nm was attributed to the (111) and (200) planes of MnO, respectively. The lattice fringe with an interplanar spacing of 0.236 nm was attributed to the (110) plane of MnCO_3_ (Figure 4d). The main existing lattice images in CMB-Ni was similar to those in B800, i.e., corresponding to MnO and MnCO_3_. However, the interplanar spacing of the lattice fringe corresponding to MnCO_3_ in CMB-Ni was 0.175 nm, which could be attributed to the (018) planes. Apart from the major phases, including MnO and MnCO_3_, the special lattice fringes of NiO was also observed in CMB-Ni. The interplanar spacing of 0.208 nm in CMB-Ni was attributed to the (012) planes of NiO (Figure 4f). These results were consistent with the XRD analysis, i.e., multi-phases existed in the materials and MnO and MnCO_3_ were major phases. The valence state of new Mn oxides formed in the thermochemical reduction process was analyzed by XPS. The spectral fitting results of the Mn(2p_3/2_) spectrogram for B800 indicated that there were three valence states, i.e., Mn^2+^, Mn^3+^ and Mn^4+^ (Appendix A), with a proportion of the corresponding peak intensity by 81.44%, 9.66% and 8.9%, respectively (Appendix A). This proved that the main oxidation state of Mn was converted from Mn^4+^ to Mn^2+^ after high temperature reduction. Meanwhile, a few high-valence Mn ions still existed, and this was consistent with the results of XRD. However, the diffraction peaks of Mn^3+^ and Mn^4+^ were not found in the XRD spectrum, which was possibly caused by the covering of the signal by background due to their very low contents. A similar valence proportion was also observed in CMB-Ni (Appendix A, Appendix A). The phases in the materials after annealing treatment mainly included MnO and MnCO_3_. Meanwhile, multi-phases were embedded in the porous carbonaceous network. The foreign oxides could act as dispersants for others in the annealing process, and inhibit the aggregation in the lithiation process and the generation of heterogeneous oxides in electrochemical cycling [10].

### 3.3. Electrochemical Determination of the Composites as Anodes for Lithium-Ion Batteries

The redox reaction of electrode materials was measured by cyclic voltammetry curves. The reduction peak near 0.1 V in the first cathode scanning for B800 presented the reduction from Mn^2+^ to Mn^0^ and the formation of a solid electrolyte interface (SEI) layer [36]. The oxidation peak at 1.25 V in the anode scanning was due to the oxidization reaction from the Mn^0^ to Mn^2+^ (Figure 5a) [1]. The redox peak intensity of cyclic voltammetry curve increased with increasing scan times, indicating the enhancement of electrochemical activity with the increase of Li ions in subsequent cycles [37].

Compared with B800, the first reduction peak of CMB-Ni shifted to 0.4 V. This indicated the occurrence of an irreversible phase change conversion, which was caused by the formation of Li_2_O and metallic Mn [1]. The new reduction peak at 0.8 V was possibly related to the presence of NiO, which corresponded to the reduction from Ni^2+^ to Ni^0^ in electrochemical cycling [34,38]. The oxidation peak at 1.25 V in the anode scan was due to the oxidation from Mn^0^ to Mn^2+^ and from Ni^0^ to Ni^2+^ (Figure 5b) [1,38]. The cyclic voltammetry curves of CMC-Ni in several cycles almost coincided, indicating the good electrochemical reversibility of the material.

The constant current charge–discharge profiles (0–3 V) of B800 with a current density of 0.1 A g^−1^ are shown in Figure 5c. As can be seen, the initial discharge and charge capacities were 756.1 and 377.7 mAh g^−1^, respectively. The Coulombic efficiency in the first cycle was 49.9%. However, the reversible discharge capacity dramatically decreased to 84.21 mAh g^−1^ after 50 cycles. The capacity degradation in the cycling process may due to the irreversible phase transition caused by the Jahn–Teller effect and the disproportionation reaction caused by Mn dissolution into the electrolyte [4]. A long and smooth discharge plateau was observed near 0.1 V in the first discharge process, which may attributable to the formation of a SEI layer and the reduction from Mn^2+^ to Mn^0^. However, the discharge plateau shifted to about 0.8 V for the second cycle due to the polarization phenomenon. The charge plateau was observed at around 1.5 V in the charge process. The large particles in the active material would be reduced to nano-metal particles in the first discharge. A high driving force and additional energy were required to supplement the interface energy and surface energy in this process, leading to the polarization phenomenon. The active materials had been fully activated in the following cycles. The requirements of energy compensation and reaction driving force greatly decreased. As a result, the discharge plateau rose, while the polarization phenomenon was obviously reduced [39,40]. The initial discharge capacities for CMB-Ni was 909.84 mAh g^−1^ at 0.1 A g^−1^ current density. The Coulombic efficiencies in the first cycle was 39.2% (Figure 5d). The additional discharge capacity of CMB-Ni compared to the theoretical capacity was possibly from the decomposition of the electrolyte in the formation of the SEI layer and the contribution of graphite [1,41,42]. Both materials, before and after incorporating, showed an irreversible loss of first discharge capacity, which was due to the decomposition of the electrolyte and the formation of the SEI layer [1,41,43]. For CMB-Ni, the polarization phenomenon was reduced. The discharge and charge plateaus were at 0.7 and 1.3 V, respectively. Note that the voltage hysteresis effect was eliminated for CMB-Ni in the long-term cycle process. The charge plateau at 1.3–1.5 V also indicated a high open circuit voltage and energy density when the materials were assembled with a specific cathode. The electrochemical testing results above proved that CMB-Ni exhibited better electrochemical properties; in particular, the capacity loss was zero after the second cycle.

The cycling and rate performances of composites obtained under different treatment conditions as anodes for lithium-ion batteries were comparatively analyzed to investigate the influence of calcination temperature and incorporated method on the electrochemical properties. In the calculation of specific capacity, the mass of the active materials reflects the total mass of amorphous carbon and metal oxides. B800 exhibited better cycling performance and rate properties than B600 and B400 (Figure 6a). In comparison with B600 and B400, B800 exhibited a much higher reversible specific capacity. This might be a consequence of the formation of MnO and C, which are the phases that lithiated well. However, rapidly capacity fade was still observed for B800. In rate performance tests of CMB-Ni, the reversible discharge specific capacity of CMB-Ni was 468.95, 390.48, 367.78, 265.44, 210.52, and 450.08 mAh g^−1^ when the current density was 0.1, 0.2, 0.5, 1, 3, and 0.1 Ag^−1^, respectively (Figure 6a), indicating that CMB-Ni had the best rate performance.

Figure 6b presents the specific capacity change of different materials within 200 cycles at a current density of 0.1 A g^−1^. The cycling capacity of the non-carbonized BMB material was the lowest. The initial discharge capacity was 335.9 mAh g^−1^ and the reversible discharge capacity after 200 cycles was only 19.39 mAh g^−1^. After calcination treatment, the cycling capacities of B400, B600 and B800 gradually increased with increasing calcination temperature. The initial discharge capacities were 569.52, 756.14 and 747.7 mAh g^−1^, respectively. After 200 cycles, the reversible discharge capacities changed to 56.14, 35.78 and 117.22 mAh g^−1^, respectively. Wherein, the initial discharge capacities of B600 and B800 were comparable to the theoretical capacity of MnO (756 mAh g^−1^). Apparently, the discrepancy in reversible cycling capacity was related to the removal effect of biological functional groups and cation vacancies in the materials. In addition, bound water components in the organic matter also caused the increase of contact resistance and the decomposition of electrolyte lithium salt, which further deteriorated the cycling performance of the battery [44]. The treatment at high temperature could also eliminate the influence of bound water components [34,35,44]. Similarly, after incorporated with metal ions, the cycling stability of CMB-Ni was greatly improved. The reversible discharge capacities of CMB-Ni at 0.1 A g^−1^ after 50 cycles was 379.29 mAh g^−1^, after 200 cycles, it can still be stable at a capacity of 356.76 mAh g^−1^. The discharge specific capacity of CMB-Ni was basically unchanged after the second cycle, and the Coulombic efficiency always stabilized at around 100%. The cycling performance for CMB-Ni improved because the multiphase metal oxides inhibited the aggregation in the lithiation process and the generation of heterogeneous oxides [9,10,45]. These results also show that the reversible charge capacities for all samples are far from theoretical capacities of MnO and NiO. The large capacity loss is mainly attributed to the irreversible processes such as electrolyte decomposition and inevitable formation of the SEI layer in the first cycle [1].

Ping et al. have previously verified the feasibility of fabricating a carbon-coating anatase anode using the cell surface display and biological template method, with a maximum specific capacity of 207 mAh g^−1^ (200 cycles) at 1C current intensity [46]. Unfortunately, the electrochemical capacity of this material seems to be difficult to improve by metal incorporation due to the intrinsic feature of the anatase and the limitation of the conventional biotemplate method for metal oxide synthesis. Moreover, several previously developed MnO-based electrode materials using biological template methods exhibited higher levels of electrochemical capacity, with maximum specific capacities at 700 mAh g^−1^ (0.1 Ag^−1^, 50 cycles) for MnO/C microspheres [1], 610 mAh g^−1^ (0.2 Ag^−1^, 60 cycles) for MnO/C microtubes [37], and 730 mAh g^−1^ (0.1 Ag^−1^, 50 cycles) [47] (Appendix A). However, these electrode materials also subjected to the capacity fade and polarization phenomenon in the cycling performance. Although the absolute specific capacity of the current composite (CMB-Ni) is not very high due to relatively low contents of Mn in materials, the significant cycle stability with nearly zero loss and polarization for CMB-Ni, prove the ability of this novel biomineralization method. It is expected that the proposed method could be extended to other biological templates to prepare new materials with improved electrochemical properties.

## 4. Conclusions

In conclusion, a facile incoporating method for Mn-based electrode materials was designed using combined bacterial surface display and biotemplate method. The mineralization ability of the engineered *P. putida* MB285 cells with surface-oriented Mn^2+^-oxidizing activity was utilized. The aggregates formed in the mineralization process were used as the biological template. Based on the settler character of biogenic Mn oxides for metal ions, the incoporating of Ni cations was realized under mild conditions. The MnO/C/NiO porous multiphasic composites were fabricated by annealing and exhibited good cycling performance in electrochemical testing, the reversible discharge capacity of CMB-Ni remained at 352.92 mAh g^−1^ after 200 cycles at 0.1 A g^−1^ current density and the coulombic efficiency was approximately 100% after the second cycle for CMB-Ni, indicating its significant potential for use in lithium-ion batteries. However, there are also problems in this method, such as the long time it took to produce the biogenic Mn oxides, but these could be improved by constructing an engineered strain with a higher Mn^2+^-oxidizing activity in further experiments. The facile biomineralization-based preparation process and novel incorporation method in this study also holds great translational promise in other similar fields.

## Data Availability

Data available in a publicly accessible repository.

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
