# Peer review of "Synthesis of MnO/C/NiO-Doped Porous Multiphasic Composites for Lithium-Ion Batteries by Biomineralized Mn Oxides from Engineered Pseudomonas putida Cells"

_nanomaterials, 2021, doi:10.3390/nano11020361_

Round 1

Reviewer 1 Report

This manuscript describes the preparation of a porous MnO/C/NiO composite (CMB-Ni) using combined unique incorporating method bacterial surface display technology and a biotemplate method. The main interest/novelty in this work is that the unique incorporating method by biomineralization was designed for nanocomposites as the anode materials for lithium-ion batteries. The manuscript is interesting; however, several points need to be addressed to further improve the quality and readability.   1. In this manuscript, pore size distribution curve of BMB based on BJH model was shown in Fig. 1(d). However, the materials which showed high performance as anode materials for Li-ion batteries are CMB-Ni, B800 and B400 rather than BMB. According to the introduction, the porous structure of MnO anode material seems to be important to prevent particle aggregation and a huge volume change in the charging-discharging process. So, I recommend that pore size distribution for CMB-Ni, B800 and B400 are described. 2. The reversible charge capacities for all samples are so small and far from theoretical capacities of MnO and NiO. The author should explain the reason for this. 3. In figure 2 (c) and 4 (a, b), there are many references of XRD patterns jumbled up together. It is very hard to distinguish the existence and ratio of components. So, these figures needs corrected.

Author Response

  1. In this manuscript, pore size distribution curve of BMB based on BJH model was shown in Fig. 1(d). However, the materials which showed high performance as anode materials for Li-ion batteries are CMB-Ni, B800 and B400 rather than BMB. According to the introduction, the porous structure of MnO anode material seems to be important to prevent particle aggregation and a huge volume change in the charging-discharging process. So, I recommend that pore size distribution for CMB-Ni, B800 and B400 are described..

We are respectful to the reviewer at this point. We have determined the pore size distribution for CMB-Ni, B800, B600 and B400. The revised Fig. S4 shows the pore size gradually increased with increasing calcination temperature.

  1. The reversible charge capacities for all samples are so small and far from theoretical capacities of MnO and NiO. The author should explain the reason for this.

We agree to the reviewer at this point. We have gave a explanation for this problem (page 12, line 449-453)

  1. In figure 2 (c) and 4 (a, b), there are many references of XRD patterns jumbled up together. It is very hard to distinguish the existence and ratio of components. So, these figures needs corrected.

The reviewer is right.We have made revision in figure 2 (c) and 4 (a, b) following the reviewer’s suggestions.

Reviewer 2 Report

The authors of the manuscript ‘Synthesis of MnO/C/NiO-doped porous multiphasic composites for lithium-ion batteries by biomineralized Mn oxides from engineered Pseudomonas putida cells’ present the synthesis of MnO/C/NiO hybrid materials using engineered cells and they also explore the potential of these materials for application in lithium-ion batteries.

In my opinion, the presented manuscript could be considered for publication after some minor revision according to the comments presented below:

  1. In Figure 1c the authors present a SEM image of the aggregates structures. The aggregates that are presented in figure 1c do not appear fully covered from the metal oxide particles. Is this SEM image of the aggregate structures formed after 16 days of incubation? Can the authors present a SEM image of higher magnification that will allow the better observation of the deposited particles in terms of size and shape?
  2. In the inset graph of figure 2d, the units do not appear very clearly.
  3. The authors present in the supporting information the TGA of B800. They should also add the TGA measurements of B600, B400 and CMB-Ni.
  4. Also, since hollow particles are formed after calcination the authors should also examine the porosity and surface area of the samples after calcination.
  5. In figure 3 the size and the quality of SEM images should be enhanced.
  6. Please the author should clarify if figure 4a presents the XRD of B800 and 4b the XRD of CMB-Ni. The same also in Figures 5a and 5b.

Author Response

  1. In Figure 1c the authors present a SEM image of the aggregates structures. The aggregates that are presented in figure 1c do not appear fully covered from the metal oxide particles. Is this SEM image of the aggregate structures formed after 16 days of incubation? Can the authors present a SEM image of higher magnification that will allow the better observation of the deposited particles in terms of size and shape?

Many thanks. The SEM image in Figure 1c showed the aggregate structures at 12 days, since it need at least 12 days for the microsphere aggregates to form. The surface of the microspheres is actually wrapped by a layer of biofilm, which makes it appear that it is not completely covered by manganese oxide and difficult to observe Mn oxides inside of the microspheres. This kind of biofilm structure is very common in Mn mineralized aggregates [1,2].

  1. Yang, W.; Zhang, Z.; Zhang, Z.; Chen, H.; Liu, J.; Ali, M.; Liu, F.; Li, L. Population structure of manganese-oxidizing bacteria in stratified soils and properties of manganese oxide aggregates under manganese-complex medium enrichment. PloS one 2013, 8, e73778, doi:10.1371/journal.pone.0073778.
  2. Zhang, Z.; Zhang, Z.; Chen, H.; Liu, J.; Liu, C.; Ni, H.; Zhao, C.; Ali, M.; Liu, F.; Li, L. Surface Mn(II) oxidation actuated by a multicopper oxidase in a soil bacterium leads to the formation of manganese oxide minerals. Sci. Rep. 2015, 5, 10895, doi:10.1038/srep10895.

  1. In the inset graph of figure 2d, the units do not appear very clearly.

The reviewer is right. We have made corrections.

  1. The authors present in the supporting information the TGA of B800. They should also add the TGA measurements of B600, B400 and CMB-Ni.

We are respectful to the reviewer at this point. Following the reviewer’s comments, we have added the TGA results of B400, B600 and CMB-Ni to Fig. S6.

  1. Also, since hollow particles are formed after calcination the authors should also examine the porosity and surface area of the samples after calcination.

We agree to the reviewer at this point. The porous structure of MnO anode material is important to prevent particle aggregation and a huge volume change in the charging-discharging process, so we determined the pore size distribution for CMB-Ni, B800, B600 and B400 and reorganized in the revised Fig. S4.

  1. In figure 3 the size and the quality of SEM images should be enhanced.

The reviewer is right. We have improve the quality of SEM images in figure 3.

  1. Please the author should clarify if figure 4a presents the XRD of B800 and 4b the XRD of CMB-Ni. The same also in Figures 5a and 5b.

The reviewer is right. We have made corrections.

Reviewer 3 Report

This manuscript is devoted to design a cation incorporating method for Mn-based electrode materials with using combined bacterial surface display and bio-template process. Authors investigated the unique incorporating method of biomineralization, in which Ni2+ ions are incorporated into the crystal structure of biogenic Mn oxides under mild conditions. The carbonization of aggregate precursor was subsequently observed at different high temperatures and carbonaceous MnO-based multiphasic nanocomposites were fabricated. It is important that they observed a further crystallization as a state evolution of the resultant materials. Finally, authors fabricated the MnO/C/NiO porous multiphasic composites by annealing, which exhibit good cycling performance in electrochemical testing, indicating its significant potential for use in lithium-ion batteries and found that electrochemical properties of materials were improved.

I have read the manuscript and found that the paper presents a new and useful research contribution to modern processes of a batteries design. However, it would be valid to mention about the place of this step on the way to the new batteries. Does the present study have a chance to be used in a future production or this route is very unsuitable, or expensive, or impossible for modern industry and this way is a standstill? I found the presentation clear and correct, and the results should be of interest for a wide audience. I also believe that the approach developed in this paper would be useful for the understanding of some other similar processes.

This paper is quite well written and also timely important. Results would be useful for many readers. The manuscript would then be suitable for publication.  I am therefore inclined to recommend for publication of the paper in “Nanomaterials”.

Author Response

  1. I have read the manuscript and found that the paper presents a new and useful research contribution to modern processes of a batteries design. However, it would be valid to mention about the place of this step on the way to the new batteries. Does the present study have a chance to be used in a future production or this route is very unsuitable, or expensive, or impossible for modern industry and this way is a standstill?

Many thanks. Following the reviewer’s comments, we have rephrased the section “Conclusion” in this revised manuscript (page 13) to describe the importance and problems of the present study to be used in a future production

Reviewer 4 Report

This study presents new valuable results on TMO-based anode materials for LIBs; it is a good attempt and interesting for readers. The manuscript is well organized and results are also clear. The paper is recommended for publication in the Nanomaterials after some revisions. 1) Novelty: authors should compare (e.g. in tabular form) the main electrochemical values of their MnO/C/NiO-doped multiphasic composite with that of other Mn-based anodes reported previously. 2) To separate Mn4+, Mn3+, and Mn2+ states in the XPS spectrum and identify their contributions, the analysis of Mn 3s is required. For this purpose please read, cite, and give some comments on the corresponding paper published by Biesinger et al. [10.1016/j.apsusc.2010.10.051] (https://xpssimplified.com/elements/manganese.php). 3) The Conclusions should involve the experimental results (numerical data), please further improve. 4) Some recent relevant papers on transition metal oxides for LIBs anodes are suggested to cite: https://doi.org/10.3390/nano9010068 https://doi.org/10.3390/nano7090252 https://doi.org/10.1016/j.jmst.2020.02.068 https://doi.org/10.1016/j.chemphys.2020.110864

Author Response

  1. Novelty: authors should compare (e.g. in tabular form) the main electrochemical values of their MnO/C/NiO-doped multiphasic composite with that of other Mn-based anodes reported previously.

Many thanks. We have compared the main electrochemical values of our study with that of other Mn-based anodes reported in previously works in page 12-13, line 454-470.

  1. To separate Mn4+, Mn3+, and Mn2+ states in the XPS spectrum and identify their contributions, the analysis of Mn 3s is required. For this purpose please read, cite, and give some comments on the corresponding paper published by Biesinger et al.[10.1016/j.apsusc.2010.10.051](https://xpssimplified.com/elements/manganese.php.

We are respectful to the reviewer at this point. However, due to the arrival of the Chinese New Year holiday, we are unable to conduct new XPS experiments. Meanwhile, the XPS fitting method used for biogenic Mn-oxide in our work is based on the previously reported literature [1], so the fitting results should be reliable. As shown in table S2, the main contribution in BMB is Mn4+, while Mn2+ play the main contribution in materials after calcination. In addition, the XRD results also consistent with the XPS results.

  1. Zhang, Z.; Zhang, Z.; Chen, H.; Liu, J.; Liu, C.; Ni, H.; Zhao, C.; Ali, M.; Liu, F.; Li, L. Surface Mn(II) oxidation actuated by a multicopper oxidase in a soil bacterium leads to the formation of manganese oxide minerals. Rep. 2015, 5, 10895, doi:10.1038/srep10895.

  1. The Conclusions should involve the experimental results (numerical data), please further improve.

The reviewer is right. We have added the experimental results to Conclusions.

  1. Some recent relevant papers on transition metal oxides for LIBs anodes are suggested to cite: https://doi.org/10.3390/nano9010068 https://doi.org/10.3390/nano7090252 https://doi.org/10.1016/j.jmst.2020.02.068 https://doi.org/10.1016/j.chemphys.2020.110864.

Many thanks. We have added these references in our manuscript.

Round 2

Reviewer 1 Report

The authors have satisfactorily answered all the questions and well revised the manuscript. So, I recommend this to be published in Nanomaterials.